# Construction of Al-Mg-Zn Interatomic Potential and the Prediction of Favored Glass Formation Compositions and Associated Driving Forces

**DOI:** 10.3390/ma15062062

**Published:** 2022-03-11

**Authors:** Bei Cai, Jiahao Li, Wensheng Lai, Jianbo Liu, Baixin Liu

**Affiliations:** Key Laboratory of Advanced Materials (MOE), School of Materials Science and Engineering, Tsinghua University, Beijing 100084, China; caib17@mails.tsinghua.edu.cn (B.C.); lijiahao@tsinghua.edu.cn (J.L.); wslai@tsinghua.edu.cn (W.L.); dmslbx@tsinghua.edu.cn (B.L.)

**Keywords:** metallic glasses, molecular dynamic, forming ability, atomic structure

## Abstract

An interatomic potential is constructed for the ternary Al-Mg-Zn system under a proposed modified tight-binding scheme, and it is verified to be realistic. Applying this ternary potential, atomistic simulations predict an intrinsic glass formation region in the composition triangle, within which the glassy alloys are more energetically favored in comparison with their solid solution counterparts. Kinetically, the amorphization driving force of each disordered state is derived to correlate the readiness of its glass-forming ability in practice; thus, an optimal stoichiometry region is pinpointed around Al_35_Mg_35_Zn_30_. Furthermore, by monitoring the structural evolution for various (Al_50_Mg_50_)_1−x_Zn_x_ (x = 30, 50, and 70 at.%) compositions, the optimized-glass-former Al_35_Mg_35_Zn_30_ is characterized by both the highest degree of icosahedral ordering and the highest phase stability among the investigated compositions. In addition, the icosahedral network in Al_35_Mg_35_Zn_30_ exhibits a much higher cross-linking degree than that in Al_25_Mg_25_Zn_50_. This suggests that there is a certain correlation between the icosahedral ordering and the larger glass-forming ability of Al_35_Mg_35_Zn_30_. Our results have significant implications in clarifying glass formation and hierarchical atomic structures, and in designing new ternary Al-Mg-Zn glassy alloys with high GFA.

## 1. Introduction

The low-density and high-strength metal alloys are of increasing interest in a wide variety of industries, including defense, sporting goods, nautical, aeronautical, automotive, and aerospace, among others [1,2,3]. The compelling need for low-density engineering materials is primarily driven by the need to reduce fuel consumption, which simultaneously reduces operational cost [4,5]. Bulk metallic glasses (BMGs), relative newcomers to the world of metal alloys, can possess a diverse array of unique properties due to their inherent glassy microstructure [6,7,8]. For example, typical low-density Al-based BMGs have occupied a special place in the material research field. Their good ductility and high specific strength, combined with excellent corrosion resistance, make them potentially desirable for integration into the marine, medical, and other fields [9,10,11,12]. Research has also shown that Mg-Zn alloys often possess good biodegradability, biocompatibility, and mechanical properties [13,14]. Mg-Zn alloys could form glassy structures over a broad composition range through rapid solidification techniques [15]. Moreover, Al and Mg are the lightweight metals available for industrial purposes, and their surface properties and strength could be significantly improved by alloying with Zn [14]. Therefore, the ternary Al-Mg-Zn system, which is a significant representative of the low-density metal alloys, is selected here for investigation. To exploit the benefits of metallic glasses (MGs), many researchers are focusing their attention on developing a model/theory capable of predicting the glass formation region (GFR) of an alloy system [16,17]. For example, Cui et al. have proposed a novel theoretical model which comprises the formation enthalpy and bond parameter function of metallic glasses and have successfully employed this model to predict the GFR of the ternary Ti-Cu-Zr system [18]. From a physical point of view, the microscopic configuration of a material, either in a disordered state or in a crystalline structure, is governed by its interatomic potential. Once the realistic potential is fitted, the favored glass formation compositions and associated driving forces for all the alloys in the system concerned could be derived through appropriate atomistic simulations.

On a fundamental level, glass-forming ability, as well as various promising mechanical and physical behaviors of MGs, are considered to depend largely upon their inherent hierarchical atomic structures [19]. In 1928, Ramsey [20] proposed that every complex structure, while seemingly random, necessarily includes ordered substructures. Nevertheless, it was not until much later that material researchers began to discover the existence of short-range orders (SROs) in MGs. Many studies have demonstrated that, as the dominant SRO in MGs, the icosahedral (ICO) and ICO-like clusters always possess good configural continuity and structural stability [21]. The formation of densely packed icosahedral ordering would increase the barrier to the formation of a crystalline structure [22,23], but would improve the GFA of a supercooled liquid. However, how these local motifs, representing various SROs, spatially distribute and interconnect with adjacent atomic clusters remains a mystery, and considerable efforts have been made to clarify a higher hierarchy of atomic configuration, i.e., the medium-range orders (MROs) [24,25,26,27,28]. Recently, studies have shown that the plastic deformation capacity of Zr-based glassy alloys could be further enhanced by tailoring the topological orders based on MRO sizes, types, and volume fractions [29]. Comparing the atomic structures in Ni_x_Zr_1−x_ and Cu_x_Zr_1−x_ MGs, it is found that relatively larger fractions of the ICO clusters were obtained in Cu_x_Zr_1−x_ MGs near Cu atoms. Meanwhile, a higher population of compacted and topologically ordered configurations and their interpenetrating connections would explain why the GFAs of Cu_x_Zr_1−x_ alloys are larger than those of Ni_x_Zr_1−x_ alloys [30]. The medium-range clusters assemble and pile up in the glassy matrix to further form complicated skeletal networks via the interpenetrating connections and the face-, edge-, and vertex-sharing linking patterns [31,32]. Recent simulations also show that the network connectivities of icosahedra correlate with the macroscopic properties (e.g., shear flows, ductility, strength, etc.) of numerous glassy materials [33,34].

In this work, based on the newly constructed Al-Mg-Zn potential, atomistic simulations are conducted to provide an increased understanding of the underlying mechanism of amorphization transition. The theoretical model proposed in this paper is expected to provide guidance for the composition design of MGs in experiments. Moreover, the atomic-level structures of (Al_50_Mg_50_)_1−x_Zn_x_ (x = 30, 50, and 70 at.%), including the optimized stoichiometry of Al_35_Mg_35_Zn_30_, were resolved by multiple analytical approaches. Furthermore, the characteristics of the hierarchical structures were quantitatively evaluated in terms of their rigidity and connectivity in order to interpret the structural origin of GFA.

## 2. Construction of Al-Mg-Zn Interatomic Potential

To pursue the MD simulations, an interatomic potential was constructed for the ternary Al-Mg-Zn system under a modified TB-SMA formalism [35]. Accordingly, the total potential energy $E_{i}$ of atom *i* can be calculated by:(1)$$E_{i}=\underset{j\neq i}{\sum }\phi \left(r_{ij}\right)+\sqrt{\underset{j\neq i}{\sum }\psi \left(r_{ij}\right)}$$

Here, $r_{ij}$ is the distance between atoms *i* and *j*. It has been shown by rule of thumb that the TB-SMA potential is applicable to hcp and fcc metals. Therefore, for the fcc-Al, hcp-Mg, and hcp-Zn, the pair function $\phi (r_{ij})$ and the density function $\psi (r_{ij})$ of Equation (1) can be written, respectively, as:(2)$$\phi \left(r_{ij}\right)=\{\begin{array}{c} A_{1}\exp\left[-P_{1}\left(\frac{r_{ij}}{r_{0}}-1\right)\right],\;\;\;\;\;r_{ij}\leq r_{m1}\\ A_{1m}\exp\left[-P_{1m}\left(\frac{r_{ij}}{r_{0}}-1\right)\right]{\left(\frac{r_{c1}}{r_{0}}-\frac{r_{ij}}{r_{0}}\right)}^{n1},\;\;\;\;\;r_{m1}<r_{ij}<r_{c1} \end{array}$$
(3)$$\Psi \left(r_{ij}\right)=\{\begin{array}{c} A_{2}\exp\left[-P_{2}\left(\frac{r_{ij}}{r_{0}}-1\right)\right],\;\;\;\;\;r_{ij}\leq r_{m2}\\ A_{2m}\exp\left[-P_{2m}\left(\frac{r_{ij}}{r_{0}}-1\right)\right]{\left(\frac{r_{c2}}{r_{0}}-\frac{r_{ij}}{r_{0}}\right)}^{n2},\;\;\;\;\;r_{m2}<r_{ij}\leq r_{c2} \end{array}$$

Here, $P_{1m}$, $A_{1m}$, $P_{1}$, $A_{1}$, and $P_{2m}$, $A_{2m}$, $P_{2}$, $A_{2}$ are another eight potential parameters to be determined by fitting. $r_{c1}$, $r_{c2}$ are the cutoff radii of the pair and density functions, and $r_{m1}$, $r_{m2}$ are the knots of the pair and density functions. As indicated in Equations (1)–(3), the pair and density functions, including their high derivatives, could smoothly and continuously converge to zero at the cutoff distances, thus obviating the force leaps, energy leaps, and the related non-physical events during the atomistic simulations [36].

The TB-SMA description of the ternary Al-Mg-Zn system is based on the constitutive unary and binary metal systems, i.e., three unary potential parameters for Al-Al, Mg-Mg, and Zn-Zn, respectively, and three binary potential parameters for Al-Mg, Al-Zn, and Mg-Zn, respectively. The Al-Al, Mg-Mg, and Al-Mg potential parameters have been fitted in Ref. [36], and thus were directly applied to this work. These unary and binary potential parameters were determined by fitting the static physical properties of pure metals and compounds obtained from experiments or *ab initio* calculations. Particularly, in the fitting process of binary potentials, compounds under various compositions or structures were utilized to ensure that the fitted potentials could truly express the interactions among atoms. Since there were insufficient experimental data for unstable elementary substances and virtual intermetallic compounds, the *ab initio* method will be applied to assist in the construction of their atomic potentials [37,38]. For the detailed *ab initio* calculation process, please refer to Ref. [36].

The potential parameters for the ternary Al-Mg-Zn system are summarized in Table 1. The related physical properties of hcp-, fcc-, and bcc-Zn derived from the TB-SMA potential, as well as calculated from *ab initio* or obtained from experiments [39,40], are listed in Table 2. It should be noted that the c/a ratio of Zn is much greater than the ideal value of 1.633; thus, the authors have decided to fit the slightly deviated c/a ratio while allowing hcp to be the lowest energy structure. The static properties of the metallic compounds in binary Al-Zn and Mg-Zn systems are also fitted by this potential and are presented in Table 3 and Table 4, respectively. Obviously, whether it is an elementary substance or intermetallic compound, their physical properties fitted by the TB-SMA potential match quite well with the results from the experiments or *ab initio* calculations, indicating that the constructed potential is reliable, and thus could be applied to the subsequent atomic simulations of the Al-Mg-Zn system.

We next verified whether this TB-SMA potential could reasonably depict the interactions between atoms under nonequilibrium conditions, i.e., derived the equation of state (EOS) from the TB-SMA potential and then compared it with the corresponding Rose equation [41]. Figure 1 shows the rose equations and EOSs for the Al-Zn and Mg-Zn compounds. It is shown that the n-body parts, pair terms, and total energies of the compounds involved in the figure are all continuous and smooth, without any discontinuities or ‘jumps’ over the entire computational range. Additionally, the EOS energy curve derived from the TB-SMA potential exhibit excellent consistency with the rose equation.

## 3. Metallic Glass Formation for the Al-Mg-Zn System

### 3.1. Evaluation of Favored Glass-Forming Compositions

The issue associated with predicting the GFR or quantitative GFA of the Al-Mg-Zn system could be addressed by applying the realistic TB-SMA potential to conduct systematic atomistic simulations, in which solid solution models are employed to compare the relative stability of the Al_x_Mg_y_Zn_1__−x__−y_ solid solutions versus their competitive amorphous counterparts. This viewpoint has been supported by many theoretical and experimental aspects [42,43,44,45].

According to the equilibrium structures of Al, Mg, and Zn, the fcc and hcp solid solution models containing 6912 atoms have been established [45]. In setting up the solid solution models, the desired solute atoms are added into the simulation models by randomly substituting a certain number of solvent atoms. Periodic boundary conditions are adopted throughout the MD simulations, and the timestep is $t=5\times 10^{-15}\;$s. The Al_x_Mg_y_Zn_1__−x__−y_ solid solution models are annealed using the Parrinello–Rahman method [46,47] at 300 K and 0 Pa for about 4 × 10^6^ MD timesteps to reach a relatively stable state (the atomic configuration and the energy of the system were almost unchanged). As a well-known feature for identifying crystalline and amorphous states, the pair-correlation functions g(r) are calculated to monitor the structural changes of the Al_x_Mg_y_Zn_1__−x__−y_ solid solutions [48].

The simulation results indicate that after the structure is completely relaxed, the Al_x_Mg_y_Zn_1__−x__−y_ models generally exhibit two different states varying the alloy composition, i.e., an amorphous state and a crystalline state. Taking Al_40_Mg_15_Zn_45_ and Al_40_Mg_25_Zn_35_ as examples, the atomic position projections, and their corresponding g(r), are shown in Figure 2 for these two alloy states. Figure 2a reveals that the g(r) curve of Al_40_Mg_15_Zn_45_ features apparent crystalline peaks. Accordingly, the atomic positions projection in Figure 2b can also visually reflect this completely ordered state. As for Al_40_Mg_25_Zn_35_ in Figure 2c, the crystalline peaks beyond the second peak have disappeared. Accordingly, the atomic position projection in Figure 2d also exhibits the typical features of short-range ordered, while long-range disordered, arrangement.

According to MD simulations, the glass formation stoichiometry diagram of the Al-Mg-Zn system is derived and exhibited in Figure 3. When the stoichiometry of the alloy falls into the gray dot area in Figure 3, the initial solid solution structures lose stability and collapse, falling into a disordered state. This gray dot area bounded by the Mg-Zn side is thus defined as the metallic glass region; whereas, when an alloy composition is located at the white dot area in Figure 3, the crystalline structure of the initial solid solution can be maintained. This white dot area is consequently classified as the crystalline region. Considering the equivalent atomic radius of Al and Zn and the completely miscible binary equilibrium phase diagram, it is almost impossible to form metallic glasses along the Al-Zn side. Until now, there is has been no report in the literature that the corresponding binary MGs could be synthesized in the Al-Mg system using any experimental methods. This is in accordance with the overall crystalline range in the Al-Mg and Al-Zn sides, as exhibited in Figure 2. As for the Mg-Zn side, only when an alloy composition falls into the central range of 20–70 at.% Zn, are the corresponding metallic glasses, rather than the competing solid solutions, more favorable to be formed. By means of first-principles molecular dynamics, Dai et al. have proposed that the intrinsic GFR of the binary Mg-Zn system to be 25–69 at.% Zn [49]. This is also extremely close to the glassy range in the Mg-Zn side.

After calculating the metallic glass region for the Al-Mg-Zn system, relevant experimental results were collected [50,51,52,53,54,55,56] and marked by the red and green triangles in Figure 3. Apparently, the compositions of these MGs obtained in the experiments all fall within the GFR predicted in this work. We can thus conclude that the intrinsic glass formation region located through MD simulations is effectively supported by the experimental observations.

### 3.2. Optimization of Glass-Forming Stoichiometries

The glass formation stoichiometry diagram in Figure 3 is not sufficient to meet the requirements of the composition design for Al_x_Mg_y_Zn_1−x__−y_ MGs, since it merely denotes the possibility of glass formation, but fails to measure the difficulty or ease of synthesizing MGs at a given composition. Therefore, it is still necessary to pinpoint the optimized compositions inside the identified GFR. From the energetic respect, the amorphization driving force [57] could be written as:(4)$$\Delta E^{am-s.s}=\Delta E^{am}-\Delta E^{s.s}$$
where $\Delta E^{am}$ is the formation energies for the glassy alloys and $\Delta E^{s.s}$ is the formation energies for the corresponding solid solutions. According to the driving force criterion, the final phase arises from the competition between the solid solution and the amorphous alloy, and the phase with the largest amorphization driving force is more likely to win this competition and achieve formation. In short, the optimized glass-forming compositions would be further distinguished by a significantly larger $\Delta E^{am-s.s}$ value.

Assuming that $E_{\mathrm{Al}}$, $E_{\mathrm{Mg}}$, and $E_{\mathrm{Zn}}$ are the experimental lattice energies [58], and MD simulation is employed to calculate the $E_{am}$ of the Al_x_Mg_y_Zn_1__−x__−y_ amorphous alloy, $\Delta E^{am}$ can be expressed by:(5)$$\Delta E^{am}=E_{am}-\left[xE_{\mathrm{Al}}+yE_{\mathrm{Mg}}+\left(1-x-y\right)E_{\mathrm{Zn}}\right]$$

In determining $\Delta E^{s.s}$, the MC method [59] is applied to calculate the $E_{\min}$ of each solid solution. $\Delta E^{s.s}$ can be determined by:(6)$$\Delta E^{s.s}=E_{\min}-\left[xE_{\mathrm{Al}}+yE_{\mathrm{Mg}}+\left(1-x-y\right)E_{\mathrm{Zn}}\right]$$

After the MD and MC calculations, the corresponding contour map of the amorphization driving force is plotted in Figure 4. It can be seen that the $\Delta E^{am-s.s}$ is always negative within the predicted region, indicating that the energy of the glassy alloy is lower than that of its solid solution counterpart; thus, the metallic glass formation is energetically favored. It should be noted that a negative amorphization driving force is a necessary, but insufficient, condition for the alloy system to be more favorable for glass formation. Further, in analyzing Figure 4, we see that the composition area marked by red dots possesses a much lower $\Delta E^{am-s.s}$. Meanwhile, the optimized composition marked by a black pentagram, i.e., Al_35_Mg_35_Zn_30_, is indicated with the lowest $\Delta E^{am-s.s}$. As stated above, the larger the formation enthalpy difference, the greater the driving force of amorphization for an alloy system. Interestingly, the experimental composition of Al_40_Mg_25_Zn_35_ collected in Figure 3 [50] is located near the pinpointed optimized composition site. It is reasonably proved that the metallic glasses near Al_35_Mg_35_Zn_30_ would be more obtainable and thermally stable.

## 4. Atomic-Level Structure of Al-Mg-Zn Metallic Glasses

### 4.1. Local Atomic Arrangements in the Short-Range

The Voronoi tessellation method was employed to designate the local short-range orders (SROs) of the MGs [60,61,62]. The distribution variations of the partial and total coordination number (CN) in the MD-derived (Al_50_Mg_50_)_100−x_Zn_x_ (x = 30, 50, and 70 at.%) MGs were investigated, as shown in Figure 5a–c. It is seen that the polyhedron with CN = 12 is dominant in both Al_35_Mg_35_Zn_30_ and Al_25_Mg_25_Zn_50_. By further inspecting Figure 5a,b, one can see that the dominating polyhedrons of the Al and Zn atoms are CN = 12, whereas the Mg atoms are mainly surrounded by CN = 14. This could be understood in terms of the atomic radii difference; the relatively larger atomic radius of Mg permits the accommodation of more atoms in its nearest-neighboring shells, and therefore favors larger CNs. However, it can be seen from Figure 5c that the dominant CNs of Al_15_Mg_15_Zn_70_ gradually increase from 12 to 13. As the Zn concentration increases to 70 at.%, the nearest-neighboring shell around each atom is inevitably forced to be packed with more small-sized Zn atoms. Therefore, more atoms would be accommodated in the nearest-neighboring distance, leading to an increase in the total CNs.

Figure 5d–f display the spectrum of the most prevalent Voronoi clusters in the (Al_50_Mg_50_)_100__−x_Zn_x_ (x = 30, 50, and 70 at.%) MGs. As exhibited in Figure 5d, the most populated clusters around Al are found to be ideal icosahedral <0,0,12,0>, followed by the icosahedra-like [63] clusters, such as <0,3,6,4>, <0,2,8,2>, and <0,1,10,2>. Moreover, the fraction of Al-centered icosahedra (~15.13%) is much higher than that centered on Mg or Zn (~0.82%, 3.57%, respectively), implying a more centralized distribution of SROs around Al. As the Zn concentration increases to 50 at.%, one observes from Figure 5e that the dominant coordination polyhedrons are indexed as <0,3,6,4>, <0,2,8,2>, <0,1,10,2>, and <0,0,12,0>, exhibiting the same prevailing local atomic clusters as the optimized-glass-former, Al_35_Mg_35_Zn_30_. Comparatively, the Voronoi cluster spectrum of Al_15_Mg_15_Zn_70_ in Figure 5f shows that the population of icosahedra-like <0,3,6,4> clusters rank topmost, whereas the fraction of the ideal icosahedra <0,0,12,0> decreases to 4.26%.

The dense clustering of icosahedra or icosahedra-like clusters would lead to the enhancement of glassy stability and the efficient filling of space. Recently, Wu et al. [64] also investigated the cluster energy distributions for different categories of SROs and found that the cluster energies of <0,3,6,4>, <0,2,8,2>, <0,1,10,2>, and <0,0,12,0> are much lower than those of other icosahedral-like clusters. The fractions of various dominant Voronoi clusters in the (Al_50_Mg_50_)_100__−x_Zn_x_ (x = 30, 50, and 70 at.%) MGs were calculated and illustrated in Figure 6. It can be seen that the fraction of the cluster <0,3,6,4> increases gradually by adding the Zn concentration, whereas the fraction of the clusters <0,2,8,2> and <0,1,10,2> decreases slowly. It is also found that, upon the addition of Zn content, the fraction of the ideal icosahedra <0,0,12,0> gradually increases in the initial stage, reaching its maximum for the optimal composition of Al_35_Mg_35_Zn_30_, and afterwards, experiences a dramatic decrease. Coincidentally, studies in Section 3.2 have indicated that the driving force for amorphization, which is an indicator used to measure the phase stability of MGs, also reaches its maximum for Al_35_Mg_35_Zn_30_. It turns out that Al_35_Mg_35_Zn_30_ is characterized by both the highest degree of icosahedral ordering and the highest phase stability, which proves that some correlation exists between the atomic structure and the glass-forming ability.

### 4.2. Structural Signature of High Glass-Forming Ability

Previous studies have shown that the full icosahedra <0,0,12,0> (FI) is energetically preferred among various local SROs, as it is absolutely made up of tetrahedra, the densest-packed cluster possible [37,65]. Figure 7 displays the distribution of FIs and their percolated networking in the simulated cells for Al_35_Mg_35_Zn_30_ and Al_25_Mg_25_Zn_50_. It can be seen that in both systems, the FI mainly overlap and interconnect with their adjacent icosahedra via the interpenetrating connections pattern, supplemented by the face-, edge-, and vertex-sharing linking patterns. However, the icosahedral network in Al_35_Mg_35_Zn_30_ is more strongly interpenetrated and developed than that in Al_25_Mg_25_Zn_50_. More extensive icosahedral networks could significantly limit the migration of shared atoms, slowing down the relevant dynamics in supercooled liquids, and subsequently decreasing the critical cooling rate of glass formation. Additionally, the distributed icosahedral network exhibits crystallographically inconsistent five-fold symmetry, which could prevent the nucleation and growth of the crystalline phase. When deviating from the optimized composition Al_35_Mg_35_Zn_30_, the icosahedral network would, to some extent, become disrupted and, according to the findings in Section 3.2, the driving force for amorphization would decrease as well. It is thus demonstrated that the formation of a higher degree of icosahedral ordering could help reduce the total energy of the system and consolidate the structural stabilities.

Of all the four categories of cluster linkages, the volume-sharing linkage is the primary interconnection pattern used in the formation of the icosahedral network. Its extensive use ensures that the system achieves a stable atomic configuration and the most favorable energetic status [66]. To quantitatively characterize how well the volume-sharing linkage works in these two icosahedral networks, their respective connectivity was further examined according to the bond number (N). In general, a higher N value demonstrates that an icosahedra bonds with more adjacent icosahedra through volume sharing [61]. It can be seen from Figure 8a,b that more icosahedra in Al_35_Mg_35_Zn_30_ are involved in the formation of the icosahedral network; when N = 0, the population of isolated clusters in Al_25_Mg_25_Zn_50_ is higher. According to the evaluation of N > 0, the icosahedral network in Al_35_Mg_35_Zn_30_ exhibits a much higher cross-linking degree than that in Al_25_Mg_25_Zn_50_. This is because when the Zn content increases to 50 at.%, the number of clusters <0,0,12,0> experiences a slight decrease and gives way to the icosahedral-like clusters, such as clusters <0,3,6,4>, <0,2,8,2>, and <0,1,10,2>, further leading to a weaker cross-linking degree of the icosahedral network.

The hierarchical atomic structures for the ternary Al-Mg-Zn MGs, including SRO, MRO, and the extended-scale structural skeleton, are illustrated in Figure 9a–c. As shown in Figure 9a, the coordination mode in the SRO is displayed by the <0,0,12,0> cluster around the Al atom. The packing mode in the MRO is shown in Figure 9b, where seven adjacent icosahedra <0,0,12,0> are collectively hybridized together in the pattern of vertex-, edge-, face-, and volume-sharing linkages to form a nano-scale super-cluster. For the connection mode of the extended-scale structural skeleton, only the center atoms of the icosahedra <0,0,12,0> are plotted as points in Figure 9c; a typical cross-linked network containing 52 icosahedra is extracted from Al_35_Mg_35_Zn_30_ MG, where the six colors correspond to various bond numbers. It is observed that the spatial distributions of the icosahedra themselves are not uniform, but rather present a serious density fluctuation. Ultimately, the densely packed spatial network that extends into the glassy matrix would deeply affect the physical and mechanical behavior of the ternary Al-Mg-Zn MGs.

## 5. Conclusions

In summary, we suggested taking the Al-Mg-Zn TB-SMA potential as the starting point, and that the computational simulations would not only predict the intrinsic GFR to be a convex region bounded by the Mg-Zn side in the composition triangle, but would also pinpoint a subregion around Al_35_Mg_35_Zn_30_ as the optimized stoichiometry area. The detailed evolution of the atomic-level structures in various (Al_50_Mg_50_)_1__−x_Zn_x_ MGs was then tracked and comprehensively characterized using Voronoi tessellation analyses. It was revealed that the dominant interconnected clusters for (Al_50_Mg_50_)_1__−x_Zn_x_ MGs are <0,3,6,4>, <0,2,8,2>, <0,1,10,2>, and <0,0,12,0>, respectively, while the population of the ideal icosahedra <0,0,12,0> increases to the maximum value at the optimized composition of Al_35_Mg_35_Zn_30_. Additionally, more interpenetrated and developed icosahedral networks in Al_35_Mg_35_Zn_30_ MG could help reduce the total energy of this system and consolidate the structural stabilities. The glass-forming abilities predicted from the structural and energetic aspects are in well accord here. Our observations establish a link between the glass formation mechanism and the hierarchical atomic structure, providing insight into how one could synthesis glassy materials with high a GFA in future experiments.

## Figures and Tables

**Figure 1 materials-15-02062-f001:**
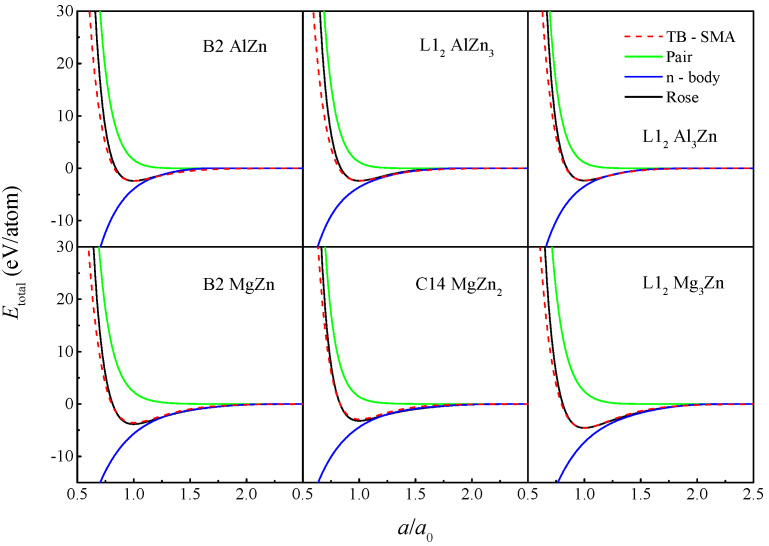
The total energies (red dash line), pair terms (green solid line), and n-body parts (blue solid line) variation with lattice constants calculated using the rose equation (black solid line), and the TB-SMA potential for compounds of AlZn, AlZn_3_, Al_3_Zn, MgZn, MgZn_2_, and Mg_3_Zn.

**Figure 2 materials-15-02062-f002:**
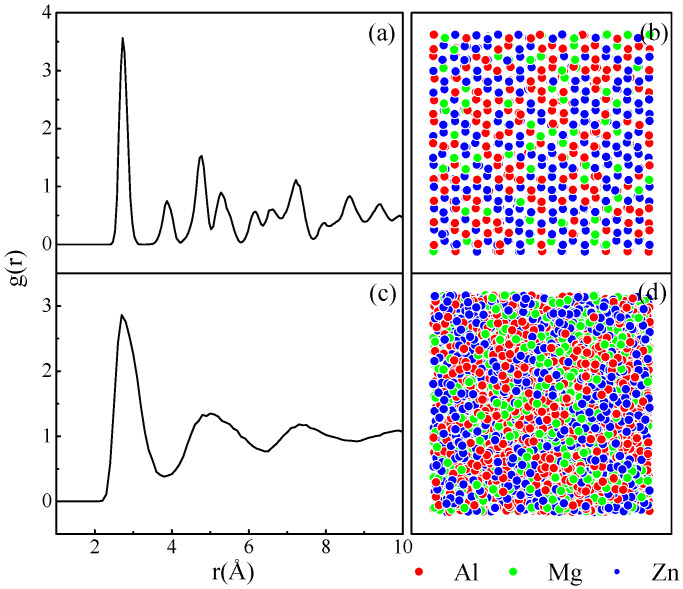
The atomic position projections and the total pair-correlation functions, g(r), (**a**,**b**) the crystalline structure (Al_40_Mg_15_Zn_45_); (**c**,**d**), the amorphous phase (Al_40_Mg_25_Zn_35_). Red, green, and blue circles represent Al, Mg, and Zn atoms, respectively.

**Figure 3 materials-15-02062-f003:**
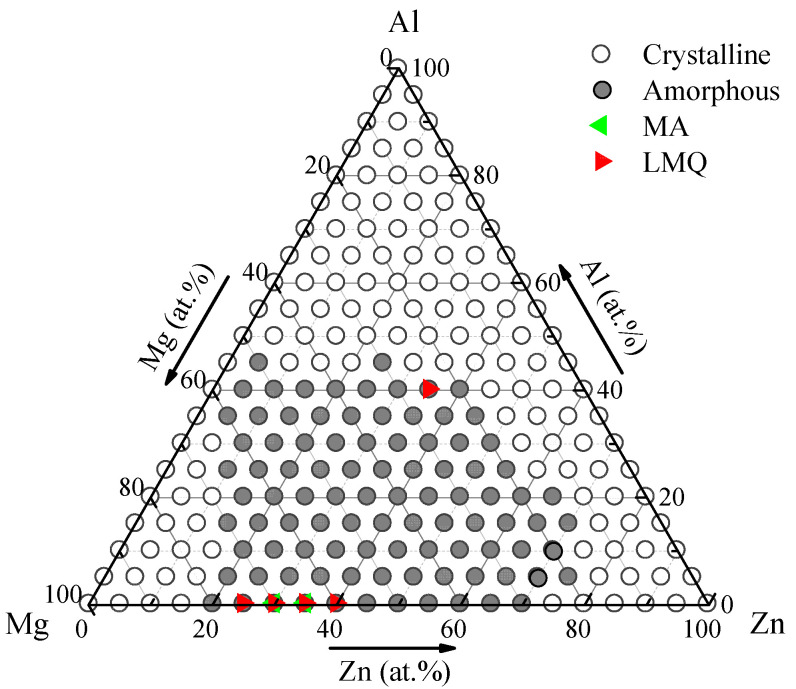
The glass formation stoichiometry diagram obtained from MD simulations at 300 K for the Al-Mg-Zn system.

**Figure 4 materials-15-02062-f004:**
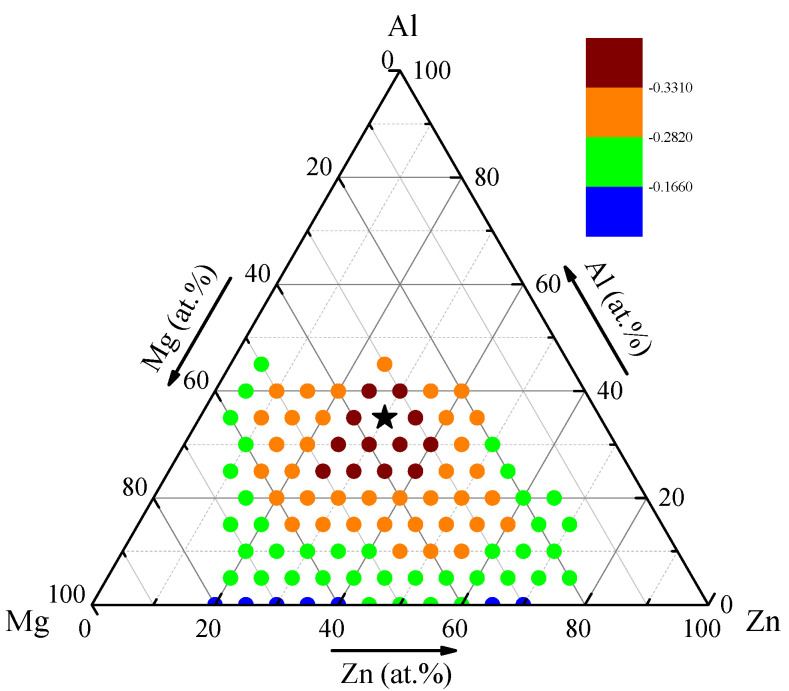
The distribution diagram of the amorphization driving force derived from the MC and MD simulations for the ternary Al-Mg-Zn system.

**Figure 5 materials-15-02062-f005:**
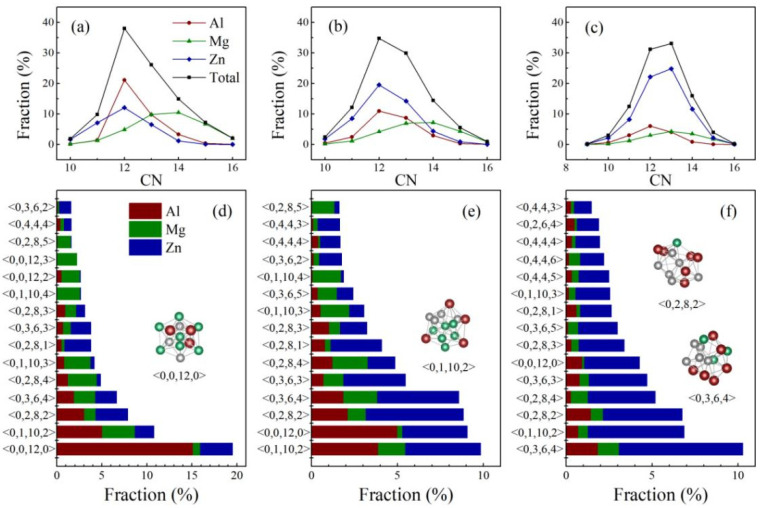
The partial and total CNs distribution of the MD-derived metallic glasses: (**a**) Al_35_Mg_35_Zn_30_, (**b**) Al_25_Mg_25_Zn_50_, (**c**) Al_15_Mg_15_Zn_70_. The spectrum of the most frequent Voronoi clusters in the MD-derived metallic glasses: (**d**) Al_35_Mg_35_Zn_30_, (**e**) Al_25_Mg_25_Zn_50_, (**f**) Al_15_Mg_15_Zn_70_.

**Figure 6 materials-15-02062-f006:**
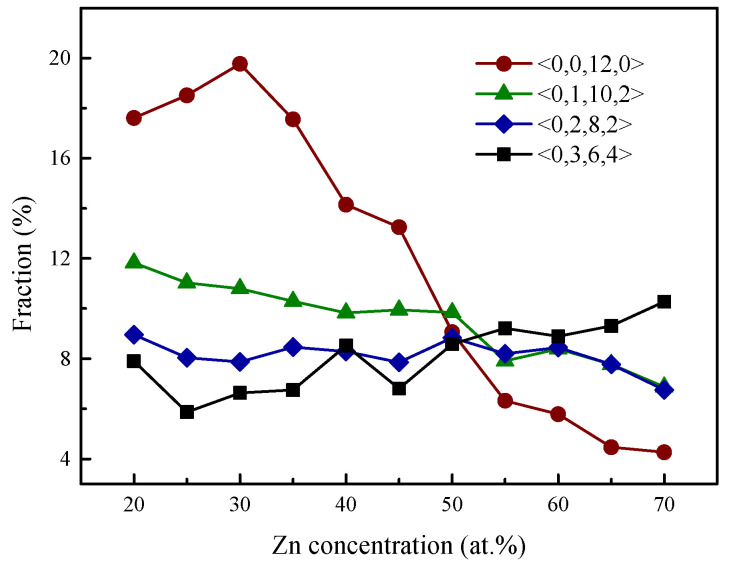
The variations in the fractions of the prevailing local clusters for (Al_50_Mg_50_)_100__−x_Zn_x_ (x = 30, 50, and 70 at.%) MGs.

**Figure 7 materials-15-02062-f007:**
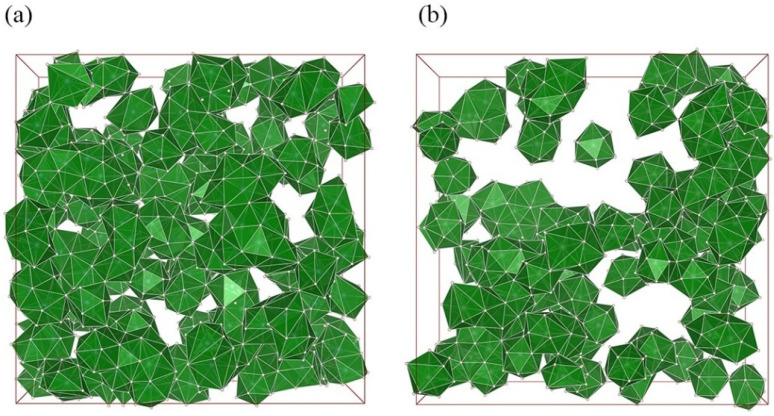
The distribution of FIs and their networking in the simulated cells for the Al_35_Mg_35_Zn_30_ and Al_25_Mg_25_Zn_50_ metallic glasses. For clarity, only the front half of the icosahedral network is shown in both (**a**,**b**).

**Figure 8 materials-15-02062-f008:**
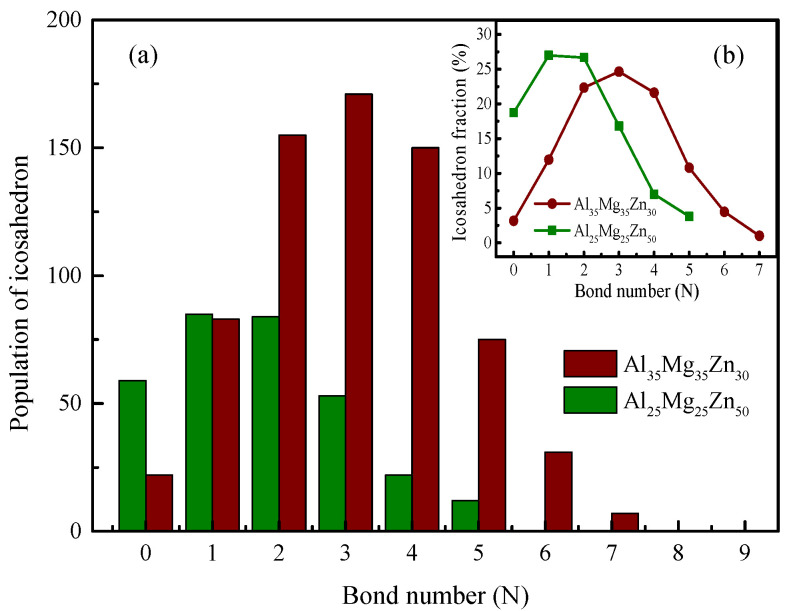
The variations in (**a**) the population, and (**b**) the corresponding fractions of different icosahedra with various bond numbers in Al_35_Mg_35_Zn_30_ and Al_25_Mg_25_Zn_50_.

**Figure 9 materials-15-02062-f009:**
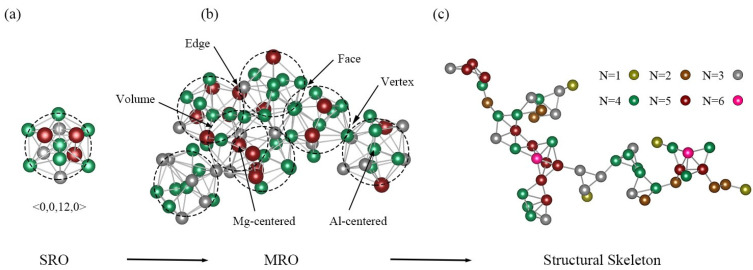
The hierarchical atomic structures for Al_35_Mg_35_Zn_30_: (**a**) The coordination mode in the local SRO is displayed by a typical Al-centered icosahedra <0,0,12,0>. (**b**) The packing mode in the MRO is characterized by a super-cluster formed among 7 neighboring FIs; each FI is highlighted with a dashed circle. The Al, Mg, and Zn atoms are colored green, red, and gray, respectively. (**c**) As for the extended scale, only the center atom of each FI is plotted as a point, and the connection mode of the structural skeleton is exhibited by a cross-linked patch containing 52 FIs. The six colors correspond to various bond numbers.

**Table 1 materials-15-02062-t001:** The potential parameters for the Al-Mg-Zn system.

Potential Parameters	Al-Al	Mg-Mg	Zn-Zn	Al-Mg	Al-Zn	Mg-Zn
*p* _1_	8.776460	10.373070	12.365126	10.238284	7.906891	9.350041
*A*_1_ (10^−19^ J)	0.402184	0.145780	0.122940	0.190712	0.518358	0.436637
*r*_m1_ (Å)	2.764394	3.522308	2.287506	2.654430	2.667892	2.590032
*n* _1_	4	4	4	4	4	4
*p* _1m_	2.558558	3.850843	5.432586	3.447699	1.328661	3.962105
*A*_1m_ (10^−19^ J)	2.917212	0.538535	3.572668	3.457161	0.505861	2.898628
*r*_c1_ (Å)	4.607023	5.487015	3.875254	4.421071	5.375373	4.800655
*p* _2_	5.249466	4.375061	6.403749	3.439822	6.359076	9.050849
*A*_2_ (10^−38^ J^2^)	4.738155	0.951887	0.600921	1.890899	3.558540	1.873409
*r*_m2_ (Å)	3.786874	2.588516	3.891120	2.636021	4.330974	3.521581
*n* _2_	5	5	5	5	5	5
*p* _2m_	0.000477	0.000378	0.000389	0.000439	0.000286	0.000486
*A*_2m_ (10^−38^ J^2^)	1.114067	1.130393	0.146081	0.441638	0.376332	6.968794
*r*_c2_ (Å)	6.515324	6.250000	6.039821	6.996006	6.538981	5.166639
*r*_0_ (Å)	2.864321	3.203567	2.751782	2.999131	2.808051	2.977674

**Table 2 materials-15-02062-t002:** The physical properties (bulk modulus (*B*_0_, Mbar), elastic constants (*C*_ij_, Mbar), cohesive energies (*E*_c_, eV), and lattice constants (*a*, *c* (Å)) of hcp-Zn, fcc-Zn, and bcc-Zn, fitted using the potential and obtained from experiments [39,40] or *ab initio* calculations.

Physical Properties	hcp-Zn		fcc-Zn		bcc-Zn	
	Fitted	Experiments	Fitted	Ab Initio	Fitted	Ab Initio
*a* or *a*, *c* (Å)	2.651, 4.614	2.665, 4.947	3.891	3.932	3.098	3.135
*E_c_* (eV/atom)	1.348	1.350	1.330	1.325	1.317	1.264
*C*_11_ (Mbar)	1.63	1.77	1.086	1.106	0.311	0.365
*C*_12_ (Mbar)	0.428	0.348	0.504	0.522	0.851	0.813
*C*_13_ (Mbar)	0.452	0.528				
*C*_33_ (Mbar)	0.403	0.685				
*C*_44_ (Mbar)	0.325	0.459	0.012	0.005	0.107	0.127
*B*_0_ (Mbar)	0.703	0.700	0.698	0.717	0.671	0.664

**Table 3 materials-15-02062-t003:** The physical properties of the Al-Zn intermetallic compounds fitted using the potential and calculated from *ab initio*.

Physical Properties	B2-AlZn		L1_2_-AlZn_3_		L1_2_-Al_3_Zn	
	Fitted	Ab Initio	Fitted	Ab Initio	Fitted	Ab Initio
*a* (Å)	3.201	3.196	3.952	3.965	4.061	4.022
*Ec* (eV/atom)	2.249	2.396	1.934	2.000	2.851	2.846
*C*_11_ (Mbar)	0.564	0.407	1.186	1.354	0.915	1.048
*C*_12_ (Mbar)	0.728	0.767	0.501	0.437	0.597	0.617
*C*_44_ (Mbar)	0.235	0.216	0.105	0.023	0.241	0.257
*B*_0_ (Mbar)	0.673	0.647	0.729	0.743	0.703	0.761

**Table 4 materials-15-02062-t004:** The physical properties of the Mg-Zn intermetallic compounds fitted using the potential (first line) and obtained from experiments or *ab initio* calculations (second line).

Physical Properties	MgZn	MgZn_3_	Mg_3_Zn	MgZn_2_
	B2	L1_2_	L1_2_	C14
*a* or *a*, *c* (Å)	3.440	4.142	4.429	5.3066, 8.2746
	3.306	4.041	4.331	5.2073, 8.5315
*E*_c_ (eV/atom)	1.508	1.459	1.493	1.522
	1.510	1.438	1.492	1.541
*B*_0_ (Mbar)	0.520	0.592	0.445	0.609
	0.501	0.595	0.413	0.653

## Data Availability

The data used to support the findings of the present study are available from the corresponding author upon request.

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
