# Peer review of "Construction of Al-Mg-Zn Interatomic Potential and the Prediction of Favored Glass Formation Compositions and Associated Driving Forces"

_materials, 2022, doi:10.3390/ma15062062_

Round 1

Reviewer 1 Report

  1. Abstract needs to be written with key results and future prospective of the field.
  2. Introduction should be rewritten with appropriate latest references.
  3. The advantage of present field must be described in the introduction.
  4. At the end of introduction, authors should clear objectives of the present research work. Also, novelty of current study is importantly needed to mention.
  5. All section must be draft one by one and make sure sequence should be logical.
  6. Results and discussion need to explore with high resolution illustration and figures.
  7. Conclusion reflects overall summary of the field with further extension and include future prospective in 2-3 lines.

Author Response

We very much appreciate the comments/suggestions from the reviewers, as those comments help us to improve the understanding of the studied issues as well as to improve the presentation of our manuscript. We accepted most of the suggestions and make corresponding modification to our manuscript.

Reviewer 2 Report

The article develops simulation techniques based on the interatomic potential (IP) of an Al-Mg-Zn alloy, and from there predicts suitable glass transition composition as well as driving forces corresponding to that suitability. Starting with the current state of MGs and their applications, the article goes on to explain why Al-Mg-Zn alloy was chosen. In addition, it is also an introduction to the influence of IP on the design of MGs. This part is well-structured and well-explained. Next is the method of performing simulations and calculations based on MD and MC to obtain the alloy composition suitable with the lowest energy formation. Later, other simulation results revealed structural change and glass transformation with the formation of a packed icosahedral network. In general, the sections of the article are organized logically and consecutively. For images, authors always explain the change in the data in great detail, followed by an explanation of why it changed. This is quite reasonable, but it is long and difficult to understand because, with so much information, it is better to summarize first, then emphasize why. Anyway, the article properly describes the phenomena, the simulation results, provides the readers with a lot of useful information. In this approach, we can see how important IP and the icosahedral network arrangement are in the construction of MGs. Thereby, demonstrating the potential of Al-Mg-Zn MGs alloy.

Author Response

I greatly appreciate the reviewers’ comments and suggestions. To be better understood by readers, we have reorganized and simplified the sections of the article to make it more logically and consecutively.

Reviewer 3 Report

Dear authors,

Thank you for presenting the manuscript 'Construction of Al-Mg-Zn interatomic potential and to predict favored glass formation compositions and associated driving forces' for consideration for publication in Materials. It deals with a highly interesting intermetallic materials system and presents some genuinely new insights into formation of intermetallic glasses. I think it is suited for publication after some minor modifications / additions:

(1) In the Abstract (line 13) you state there is a region within which the glass formation is 'energetically favored'. This is somewhat awkwardly formulated - all glasses are thermodynamically unstable.

(2) The degree of icosahedral ordering is taken as a measure for the ability of a system to form glasses. I don't, however, believe this to be a good criterion - you could also take it for a measure of a system to form quasi-crystalline phases. Could you please comment on this a bit more in detail?

(3) Please change all 'icosahedrons' into 'icosahedra'.

(4) Can you, based on your methods, give estimates of the energy barrier stabilising the glass system from crystallisation? This would be highly interesting and I see a high potential to do so by your methods.

(5) I would like to see a correlation of your glass formation predictions with experimental work, and of course this is out of the scope of this manuscript, but I hope it will be topic of future work.

Sincere greetings

Author Response

(The authors gave the same response as above.)

Round 2

Reviewer 1 Report

Proofreading is required.